# High Cycle Fatigue Behaviour of 316L Stainless Steel Produced via Selective Laser Melting Method and Post Processed by Hot Rotary Swaging

**DOI:** 10.3390/ma16093400

**Published:** 2023-04-26

**Authors:** Petr Opěla, Marek Benč, Stepan Kolomy, Zdeněk Jakůbek, Denisa Beranová

**Affiliations:** 1Faculty of Materials Science and Technology, VSB–Technical University of Ostrava, 17. Listopadu 2172/15, 70800 Ostrava, Czech Republic; petr.opela@vsb.cz; 2Faculty of Mechanical Engineering, Brno University of Technology, 61600 Brno, Czech Republic; stepan.kolomy@vutbr.cz; 3Institute of Physics of Materials, Czech Academy of Sciences, Žižkova 22, 61600 Brno, Czech Republic

**Keywords:** 316L steel, selective laser melting, hot compression testing, hot rotary swaging, high cycle fatigue, microstructure

## Abstract

This paper deals with a study of additively manufactured (by the Selective Laser Melting, SLM, method) and conventionally produced AISI 316L stainless steel and their comparison. With the intention to enhance the performance of the workpieces, each material was post-processed via hot rotary swaging under a temperature of 900 °C. The samples of each particular material were analysed regarding porosity, microhardness, high cycle fatigue, and microstructure. The obtained data has shown a significant reduction in the residual porosity and the microhardness increase to 310 HV in the sample after the hot rotary swaging. Based on the acquired data, the sample produced via SLM and post-processed by hot rotary swaging featured higher fatigue resistance compared to conventionally produced samples where the stress was set to 540 MPa. The structure of the printed samples changed from the characteristic melting pools to a structure with a lower average grain size accompanied by a decrease of a high fraction of high-angle grain boundaries and higher geometrically necessary dislocation density. Specifically, the grain size decreased from the average diameters of more than 20 µm to 3.9 µm and 4.1 µm for the SLM and conventionally prepared samples, respectively. In addition, the presented research has brought in the material constants of the Hensel-Spittel formula adapted to predict the hot flow stress evolution of the studied steel with respect to its 3D printed state.

## 1. Introduction

The additive manufacturing (AM) process is a revolutionary technology that attracts attention due to its ability to produce final products. The process of the AM can be used to build three-dimensional complex parts based on computer-aided design (CAD) data which have further utilization in automotive, nuclear, or medical industries [1,2,3]. A micro welding process is close to the AM process; in both cases, laser power is used for melting a metal powder. Increasing laser power can impart a decrease in the number of cracks and cracking width, which can affect the mechanical properties of a welded part [4]. Common AM technologies are direct energy deposition and laser powder bed fusion (LPBF). The LPBF technology consists of Direct Metal Laser Sintering (DMLS), Electron Beam Melting (EBM), Selective Laser Sintering (SLS), and Selective Laser Melting (SLM) [5,6,7]. Using the SLM, the metal powder is deposited on the building bed, where the powder is gradually melted layer by layer using a powerful laser [8]. SLM can be used for a wide range of metallic materials, such as maraging steels, titanium alloys, or aluminium alloys [9,10], and also the AISI 316L stainless steel featuring favourable mechanical properties, weldability, corrosion resistance, and also biocompatibility [11,12,13].

AM materials feature different structures, porosity, or mechanical properties depending on the process parameters, printing strategy, or chemical heat treatment [14,15]. Numerous papers focused on the study of the final structure with different process parameters or printing strategies have been published. For example, studies [16,17,18] investigated the influence of different hatching strategies on the final structure of nickel-based alloys. Wan et al. [19] found that the control of a heat flux direction between the successive layers using a different scanning strategy forms such different structures within additively manufactured Inconel 718. Liu et al. [20] concluded that using an optimized scanning strategy, Ti-6Al-4V alloy manufactured by LPBF exhibited a low variation in mechanical properties, especially in the ultimate tensile strength. Nong et al. [21] investigated the effect of the scanning strategy on 15-5PH stainless steel; using the island scanning strategy, high-densification samples with fine grain structures were obtained. Song et al. [22] dealt with the effect of the scanning strategies on 316L stainless steel and found that using the scanning strategy without a rotation between the successive layers caused the growth of columnar epitaxial grains, but with a rotation between the layers, the columnar growth of grains was not observed, which contributed to the grain refinement.

Mechanical properties can be improved by applying appropriate heat treatment, Hot Isostatic Pressing (HIP), or plastic deformation processing [23,24]. Researchers [25,26] evaluated the effects of heat treatments on sintered nickel-based alloys. They concluded that the microhardness increased after aging for 2 h and decreased after a prolonged aging time at a higher temperature. Liu et al. [27] observed an effect of solution-aging treatments on the mechanical properties of AM Ti-6Al-4V alloy and revealed that the solution-aging treatments increased the tensile strength but decreased ductility. Chen et al. [28] investigated the effects of heat treatment on the mechanical properties of AM AISI H13 tool steel. They concluded that the heat treatment at 550 °C brought in the highest microhardness, which was caused by needle-like carbide precipitates. Shin et al. [29] examined the effect of the heat treatment on the mechanical properties of 316L steel and found that the heat treatment improved elongation, while the surface hardness and tensile strength decreased. Liang et al. [30] found that the structure after HIP featured a series of typical columnar γ grains that grew along the building direction. Qu et al. [31] revealed that the ultimate tensile strength and yield strength of a Ti-6Al-4V alloy sample decreased with a higher annealing temperature during HIP. Topping et al. [32] showed that, despite the grain growth during HIP, an ultrafine grain structure within an aluminum alloy was retained, which consequently increased strength compared to the cast alloy. Liverani et al. [33] documented that post-processing by HIP increased the relative density of AM 316L steel to above 99%.

Residual stress, porosity, structure, and mechanical properties after SLM can be improved via an intensive plastic deformation, e.g., by rotary swaging [34]. It is a manufacturing process during which the cross-sectional area of the workpiece is reduced and its length is increased by repeated radial blows with a minimum of two rotating dies. The imposed strain via hot rotary swaging can range from relatively low values, similar to e.g., rolling [35], to very high values, such as during severe plastic deformation [36,37,38]. The method allows to post-process of printed SLM parts to create functional finished parts with no need for further processing. For example, Estrin et al. [39] examined the effects of rotary swaging on an Mg-Al-Zn alloy and concluded that the processing increased the strength and ductility of the alloy remarkably. Macháčková et al. [40] performed thorough numerical and experimental investigations of the effects of a processing temperature applied during rotary swaging on formability and a structure of a powder-based tungsten heavy alloy, while Kunčická et al. [41] studied the effects of intensive deformation on grains orientations and residual stress within clad composites and confirmed that optimized deformation processing provides favourable bonding of the individual layers. 

Components, such as shafts, gearboxes, etc. should operate under dynamic loading for a long time, but defects and pores situated in a surface layer can cause an initiation of micro cracks, which leads to the destruction of the part exposed to high cycle fatigue or dynamic load [42]. Researchers [43,44] examined low and high cycle fatigue resistance of Ti-6Al-4V alloy produced via SLM and conventionally; the SLM part featured a lower fatigue strength. Studies [45,46,47] examined a dependence on the microstructure of AM 316L steel and a fatigue performance; the AM part featured a lower fatigue resistance than a conventionally produced part. Solberg et al. [48] found that a crack within AM 316L steel was initiated from defects in the surface, while under a higher stress value, the crack was nucleated from internal defects. Investigating the performance of AM materials can also advantageously be put in correlation with computer tomography (CT). Researchers [49,50] investigated the effect of the magnitude and variance of fatigue performance on AM materials and concluded that the fatigue curves of specimens with different initial defect sizes are closely related.

This paper deals with a comparison of AISI 316L stainless steel produced conventionally and via SLM. The comparison involves microhardness, high cycle fatigue, and microstructure. Currently, there are not many papers dealing with the testing of the material fatigue resistance enhanced via rotary swaging. Samples were subjected to hot rotary swaging under a temperature of 900 °C and subsequently tested under high cycle fatigue. The comparison between additively and conventionally manufactured samples subjected to post-processing has been described in detail. The microstructure was examined using electron backscatter diffraction (EBSD). The evaluation involved grain size, texture, a fraction of high-angle grain boundaries, and geometrically necessary dislocations.

## 2. Materials and Methods

### 2.1. Examined Material

AISI 316L stainless steel was analysed in two states, i.e., conventionally, and additively processed samples. The conventional material was prepared as a cast and rolled product. Additively manufactured samples were produced using AM400 3D printer (Renishaw plc., Wootton-under-Edge, UK). The chemical composition of the virgin 316L powder is presented in Table 1. The size of powder particles was within a range of 15–45 µm. The process parameters used for the SLM process are depicted in Table 2. The building process was protected by an inert atmosphere provided by argon gas with a purity of 99.998%. The shape of the as-fabricated samples was a cylinder with a diameter of 30 mm and a length of 60 mm (see Figure 1a). The laser direction was oriented from left to right with a focus size of 70 µm, while the inert gas was flown from the opposite site. The meander hatching pattern (see Figure 1b) was applied for the fabrication of the samples which were printed in the vertical position. This strategy provides sufficient melting of the powder and a formation of a structure with favourably low porosity [51]. The energy density used for the fabrication was calculated according to Equation (1) [52]:(1)$$E=\frac{P}{t\cdot h\cdot v}$$
where *E* is a laser energy density (J·mm^−3^), *P* is a laser power (W), *t* is a layer thickness (mm), *h* is a hatch spacing (mm), and *v* is a scanning speed (mm·s^−1^).

### 2.2. Hot Compression Testing

Since the samples of the examined material are intended to undergo a hot rotary swaging procedure, it is necessary to know their behavior under hot deformation conditions—specifically the flow stress evolution. The hot deformation behavior of the conventionally prepared AISI 316L stainless steel has been known already—see flow curves in [53]. Nevertheless, with respect to the additively manufactured samples, one can assume that the applied 3D printing technology will naturally lead to a different hot deformation behavior. Since this behavior has not been known yet, plastometric tests were performed in order to establish the corresponding flow stress courses. To do so, the additively prepared samples of the studied steel have been machined to gain 16 cylindrical samples (a diameter of 10 mm, and a length of 15 mm). The prepared samples were then subjected to a series of uniaxial hot compression tests (performed on the Gleeble testing machine, (Gleeble, Poestenkill, UK) under four deformation temperatures, specifically 900, 1000, 1100, and 1250 °C. Each examined temperature level was combined with four strain rates (0.1, 1, 10, and 100 s^−1^) under the following accompanying settings: sample heating performed directly up to a deformation temperature (the heating rate of 10 °C·s^−1^ provided by direct electric resistance heating), dwell-time on a temperature before the deformation was equal to 300 s, the true strain value set to 1.0, the testing chamber held under vacuum, the sample-anvil interface treated by tantalum foils in a combination with a nickel-based high-temperature grease. The described procedure enabled to gain 16 hot flow curves covering the flow stress evolution of the examined material under a wide range of thermomechanical circumstances.

The acquired flow curve dataset was subsequently processed via a regression analysis in order to gain a functional relationship between the deformation temperature, *T* (°C), strain rate, $\dot{\varepsilon }\;$(s^−1^), true strain, *ε* (-) and true flow stress, *σ* (MPa). Considering various approaches enabling assembly of the required functional relationship, see e.g., [54], the well-known Hensel-Spittel formula was selected regarding its implementation in the FEM software Forge—see the following relation [55]:(2)$$\sigma =A\cdot e^{m_{1}\times T}\times \varepsilon^{m_{2}}\times {\dot{\varepsilon }}^{m_{3}}\times e^{m_{4}/\varepsilon }\times {\left(1+\varepsilon \right)}^{m_{5}\times T}\times e^{m_{7}\cdot \varepsilon }\times {\dot{\varepsilon }}^{m_{8}\times T}\times T^{m_{9}}$$
where *A*, *m*_1_, *m*_2_, *m*_3_, *m*_4_, *m*_5_, *m*_7_, *m*_8_, and *m*_9_ represent material constants. The determination of these constants was performed via a nonlinear regression analysis by applying the Levenberg-Marquardt nonlinear least square optimization algorithm [56,57] in the Octave-4.2.1 software applying the optim-1.5.2 package. 

### 2.3. Hot Rotary Swaging

Both materials were processed by the hot rotary swaging technology. The samples were heated to a temperature of 900 °C using an electric resistance furnace. Each sample was placed into a fixture and subjected to the rotary swaging process. The cylindrical sample was plastically deformed by a set of four dies which rotated around the simultaneously rotating cylindrical sample until the final diameter reduction (10 mm). A comparison of each sample state is shown in Table 3, i.e., the 3D printed sample (SLM), the conventionally manufactured sample (CON), the 3D printed sample plus hot rotary swaging (SLM + RS), and the conventionally manufactured sample plus hot rotary swaging (CON + RS) was performed. 

### 2.4. High Cycle Fatigue Testing

The rods after post-processing were prepared as the samples (SLM + RS and CON + RS) used for further analysis. These samples were observed in regard to mechanical properties (especially microhardness), high cycle fatigue, and microstructure. High-cycle fatigue tests were performed using the universal testing machine Instron E10000 (Instron Norwood, MA, U.S.) with linear motor technology. Tests were performed in a stress-driven regime, with a load ratio R= –1 and a testing frequency of 15 kHz. Tests were carried out at an ambient temperature of 23° ± 2 °C. A total of 14 samples (10 samples from the conventional production and 4 fabricated via SLM) prepared by the hot rotary swaging post-process were exposed to a high cycle fatigue test containing 2 × 10^6^ loading cycles.

### 2.5. Microstructure Observation and Microhardness Measurement

The samples for microstructure analyses were prepared in the transverse (the XY plane) and the longitudinal (the YZ plane) directions. The samples were subsequently cast in Bakelite and then mechanically ground using SiC emery papers. The final surface of the samples was prepared via the electropolishing process. The microstructure was observed via scanning electron microscopy (SEM) using the electron backscatter diffraction (EBSD) method. The observation was performed using the microscope Tescan Lyra 3 XMU FEG/SEMxFIB. The EBSD maps were obtained using a Symmetry EBSD detector (Tescan Orsay Holding a.s., Brno, Czech Republic). The acquired maps were analysed via Aztec Crystal software (Oxford Instruments, Abingdon, UK). The samples were scanned in 100 × 100 µm with the scanning step of 0.1 µm. The samples subjected to hot rotary swaging were scanned with a higher magnification in the area of 50 × 50 µm. The microhardness was measured in a transverse (the XY plane) and longitudinal (the YZ plane) direction via Zwick Roell equipment (Zwick Roell CZ s.r.o., Brno, Czech Republic). The distance between each indention was 1 mm corresponding to the applied load of 200 g (HV 0.2) and a dwell time of 15 s. 

## 3. Results and Discussion 

### 3.1. Hot Flow Stress Evolution

The material constants of the Hensel-Spittel Formula (2) calculated to fit the experimental flow curve dataset of the studied steel are shown in Table 4. The good regression fit of the assembled flow stress model can be documented by a low Root Mean Squared Error value, *RMSE* = 9.52 MPa, and a high value of the Pearson correlation coefficient, *R* = 0.9944.

Figure 2 graphically compares the experimental and calculated flow stress courses in the form of flow curves. At first sight, it is evident that the flow stress course takes the expected trends, i.e., the flow stress growth with a decreasing temperature level and an increasing strain rate. One can see the Hensel-Spittel model is able to fit the experimental data well—though with an exception in the case of the temperature level of 1100 °C in combination with the strain rate of 0.1 s^−1^—see Figure 2a. However, it is noticeable that the course of the experimental curve under these conditions is out of the trend observed for the surrounding curves—the flow stress decrease seems to be somewhat overestimated. So, the overall performance of the applied model can be evaluated as high enough and the corresponding material constants in Table 4 can be used to predict flow stress courses under non-experimentally examined conditions. 

### 3.2. Porosity Evaluation

The printing parameters (a laser power of 200 W, a layer thickness of 0.05 mm, a hatch spacing of 0.06 mm, and a scanning speed of 650 mm·s^−1^) used during the building process influenced a formation of a structure that featured defects such as pores and voids (see Figure 3a). The defect can be caused by the protecting gas which was entrapped in the material during the deposition of the individual layers [58,59]. The residual porosity was evaluated in the middle height plane of the samples (see Table 3) due to the expectation of the highest porosity [41]. The ImageJ (version 1.53t) program was used for the evaluations of the residual porosity. The porosity evaluation was performed on the polished light optical microscopy image with the methodology according to Grove and Jerram [60]. The structure of the SLM and SLM + RS sample exhibited residual porosity of 1.836% and 0.051%. A significant reduction in the residual porosity was observed after the hot rotary swaging post-process (see Figure 3b) for every tested sample listed in Table 3. It can be concluded that the lower residual porosity can increase fatigue resistance due to the reduction in potential stress concentrators. 

### 3.3. Microhardness Evaluation 

The dependence of the Vickers microhardness on a distance across the sample is presented in Figure 4. The microhardness of the samples corresponding with Table 3 was measured across each sample in the transverse (the XY plane, see Figure 4a) and in the longitudinal (the YZ plane, see Figure 4b) direction. The microhardness showed a similar trend for both the materials in the transverse direction as well as in the longitudinal direction. The lowest microhardness was measured in the vicinity of the border (230 HV) of the SLM sample in the XY plane, while the highest value (250 HV) was observed in the centre of the sample. The course of the measured microhardness exhibited such a wavy trend. These waves can be related to different regions appearing across the structure formed during the SLM process. The variation of the microhardness (an increase and a decrease) corresponds with two distinct areas with a different etching contract, i.e., darker areas of the tempered martensite and brighter areas presenting less extensive tempering [61]. The CON sample in the XY plane features a different trend, where the highest microhardness (280 HV) was observed in the vicinity of the sample surface, while the lowest (221 HV) was presented in the centre of the sample. This fact corresponds with the post-process rolling treatment (during the production), which affected mostly the surface area of the sample. The SLM sample in the YZ plane featured a similar wavy trend of microhardness as in the XY plane. The existence of two distinct areas (the melting pools structure with the different etching contrast is described in detail in the section: Microstructure observation) with a different microhardness corresponded to the following fact: the high melting temperature imposed heat into each deposited layer, which could cause partial tempering of the previously melted layer [62]. The post-process of hot rotary swaging caused an increase in microhardness in both materials. The average value calculated from all indentions exhibited 310 ± 5 HV and 311 ± 5 HV in the case of SLM + RS and CON + RS in both directions. It can be observed that the difference in the material microhardness (observed near the sample surface) diminished after post-processing. In other words, the hot rotary swaging affected the structure, which resulted in homogeneous microhardness (no wavy course has been observed) across the sample. A similar homogenization of a structure was observed by Chen et al. [63]. They found out that with the increasing post-heat treatment temperature the microstructure of the material gradually homogenized.

### 3.4. High Cycle Fatigue

The fatigue resistance of additively manufactured samples depends on microstructural features (especially on grain size, porosity, and high-angle grain boundaries) and surface quality (in the study, the samples were first machined to a diameter of 3.5 mm at the centre of the sample and subsequently polished with 0.2 µm paste) related to under-surface defects and microcracks. The features can be affected by post-processing using machining or hot rotary swaging. Each tested specimen was loaded in a tension-compression (the stress ratio, R = σ_max_/σ_min_, was −1) cycle with a resonance frequency of 15 kHz. The material was exposed to the maximum stress within the range of 420–560 MPa. Four individual SLM samples subjected to the rotary swaging were tested under the stress of 460 MPa and 560 MPa (2 samples for each stress). 

The acquired data from the high cycle fatigue tests for the conventionally and SLM post-processed samples are listed in Table 5. The SLM + RS1 and SLM + RS4 samples were exposed to the lowest and highest applied load of 460 MPa and 540 MPa. The samples withstood 146,542 cycles to failure in the case of the SLM + RS1 and 469,289 cycles in the case of the SLM + RS4. In the low-stress regime, the cycle to failure of both materials exhibited a very similar level but a larger difference was observed at a higher applied stress. Based on the data listed in Table 5 it is evident that the SLM post-processed samples exhibited a higher cycle to fracture compared to the conventional post-processed samples. In the case of the applied stress of 540 MPa the samples (SLM + RS and CON + RS) featured a lower cycle to failure in comparison to the stress of 460 MPa. In other words, the SLM + RS samples exhibited a higher fatigue resistance compared to the CON + RS sample under the stress of 540 MPa. This was more than doubled in the case of 540 MPa and 2.02% in 460 MPa. It can be said that under a lower stress (460 MPa) the fatigue resistance was very similar, but under a higher stress a higher difference (more than two times) was observed.

The results of the high cycle fatigue for both materials are presented in Figure 5 in a logarithmic scale, which shows that the cycles to failure increased with a decreasing applied load. Similar results as presented in this study were obtained in the study [64], which compared a coarse grain structure material and a material subjected to a surface mechanical rolling treatment. They found out that the fatigue resistance increased after the exposition to the post-process of the surface mechanical rolling treatment. The obtained results from this study were compared with other studies i.e., Voloskov et al. [65], Spierings et al. [66], Leuders et al. [67], and Xiong et al. [68] which are also included in Figure 5. In the study [68], the solution-annealed condition exhibited the lowest performance indicated by the lowest fatigue life, followed by the as-build condition (described in [67]), which referred to the slightly higher fatigue life compared to the solution-annealed sample. The very high cycle fatigue resistance of SLM 316L was increased in the case of the cold-worked specimens (marked in [68]). The results shown in Figure 5 featured discrepancies in comparison to the results described in the previous studies, which could be affected by the different dimensions of the tested samples. Due to the different dimensions of the samples, the relation between a crack initiation life and a crack growth life could be changed [67].

According to [69], the fatigue behaviour of the 316L alloy is primarily determined by the monotonic strength and not by the process-induced defects within the range of the relatively high stress. When the specimen is exposed to lower stress (very high cycle numbers), small pores can affect the crack initiation and cause damage. The positive effect of rotary swaging is visible in this study since the regression lines for both of the rotary-swaged materials were significantly above the lines of the other research. 

Post-processing via rotary swaging introduced densification of the material, thus resulting in higher fatigue resistance.

The samples exposed to high cycle fatigue were analysed via SEM to observe the detailed area of the rupture. These samples, which were chosen, exhibited the highest difference in obtained cycles to failure in the case of the stress of 460 MPa. The fracture surfaces represented the CON + RS3 (N = 460,047 cycles before rupture) and the SLM + RS4 (N = 469,289 cycles before rupture) samples tested in the high cycle fatigue regime under the stress level of 460 MPa. The fracture surface (see Figure 6a,d) consisted of a fatigue crack propagation zone (A) and a final rupture zone (B). The propagation zone nucleated from the outer surface continuing towards the sample axis. The direction of the crack propagation is indicated by the yellow arrows in Figure 6a,d. The rupture surface (depicted in Figure 6b,e) is characterized by a complicated viscous relief of the fracture with fine dimples (especially visible in Figure 6f). The rupture surface exhibited a similar relief as obtained by Fu et al. [70] during the slow strain rate tensile tests. The yellow arrows depicted in Figure 6b,e indicate the crack initiation and propagation mechanisms from manufacturing defects. The visible defects (unmelted particles marked with the yellow circle) are depicted in Figure 6c,f). They are detected at the nucleation area where the crack propagation increased. Both the observed samples featured no significant difference as regards the ruptured surfaces of the samples due to the similar number of absorbed cycles before the rupture.

### 3.5. Microstructure Observation 

A detailed optical microscopy analysis of the SLM sample (corresponding to Table 3) was further carried out. Figure 7 shows LOM images taken of the as-fabricated sample. The sample exhibits a typical structure (characteristic melt pools featuring individual grains, see Figure 7a) formed during the melting process. The structure exhibits a region of the rapid cooling effect (Q = Quenched region marked in Figure 7b) and a region of partial tempering (T = Tempered region) described in Section 3.3. The quenched regions correspond to the characteristic martensitic laths. This structural feature has most probably contributed to a difference in the measured microhardness (such a wavy trend) achieved for this sample.

The integrated pictures depicted in Figure 8 represent the grain structures (the grain orientations are differentiated in colours) from the centre of the specimen to the border of the surface. The images, which were captured at 200× magnification, represent the horizontal (the XY plane) as well as the vertical (the YZ plane) direction of the sample, corresponding with the electron backscatter diffraction EBSD maps. Each plane was observed in two regions, i.e., the area at the border (an interface between the sample and the bakelite) and the area in the centre of the sample. The coloured grains correspond to their individual orientation according to the Miller indices which are shown in the orientation image maps (OIM, see the triangle in the corner of each figure). The microstructures of the samples in each region (SLM and CON listed in Table 3) are presented in Figure 9a–h. 

The SLM material exhibited a characteristic melting pool structure formed during the 3D printing process containing a mixture of large and small grains which featured no tendency to form into a certain preferential orientation. Larger grains were observed at the border of the sample (see Figure 8a,c), while smaller grains were preferentially formed in the centre of the sample in the XY as well as in the YZ plane (see Figure 8b,d). The average grain size (a summary of the average grain size for each sample is depicted in Figure 8i) was measured as max-ferret diameter (considered as a maximum distance between two points within a single grain). The average grain size in the SLM sample was slightly differentiated in the observed XY plane and the YZ plane, but in the CON sample, a higher difference in the average grain size was observed in both the XY and YZ planes. Within the XY direction, the grain size exhibited lower values in all cases compared to the YZ plane. Each sample exhibited a higher average grain size at the border of the sample within both the XY and YZ directions. As to the CON sample, the difference between the average grain size in the centre and at the border of the sample exhibited 11.8µm in the XY plane (see Figure 8e,f) and 15.2 µm in the YZ plane (see Figure 8g,h). This was a 102% and 115% increase in the horizontal and vertical directions. The rolling applied during the production of the CON sample resulted in lower average grain size in both the regions and planes. 

The low angle grain boundary (LAGB, captured in a range of 5–15°) and the high angle grain boundary (HAGB, a threshold was considered to be 15°) were evaluated for each sample. The SLM sample featured the lower HAGB fraction in the centre of the sample in the XY (the observed difference was 6.6%) and YZ (the observed difference was 1.1%) plane compared to the border area, while the CON sample exhibited the higher observed HAGB fraction in the centre compared to the border. The difference observed in the XY plane featured an increase of 2.6% and in the YZ plane an increase of 4.3%. Based on the obtained data, the structure with the lowest average grain size exhibited the highest HAGB fraction.

EBSD orientation image maps for the samples exposed to hot rotary swaging (SLM + RS) and (CON + RS) are presented in Figure 9. The hot rotary swaging caused significant structure and geometrical changes resulting in a decrease in the average grain size and the proportion of HAGB fraction. A fully recrystallized structure of the 316L stainless steel was formed during post-processing by hot rotary swaging. The structure consists of small epitaxial grains formed in the horizontal direction, while the structure in the vertical direction exhibits grains with their longer axis formed predominantly parallel to the building direction. This phenomenon corresponds with the process of hot rotary swaging, where the sample was processed with four dies rotating around the sample. The crystallographic texture in the border areas of the XY and YZ planes of the SLM + RS was evaluated via the inverse pole figures (IPFs, depicted in Figure 9c,f). The structure exhibited a relatively strong tendency to form the <001> axis (‖z) and <101> (‖z) preferential orientations in the horizontal and vertical directions.

The average grain size (the average grain size summary for all samples is depicted in Figure 9k) considering the SLM + RS and CON + RS samples decreased in the XY and YZ planes within the observed regions. Generally, the structure exhibited a lower grain size in the horizontal direction compared to the vertical direction. The differences in the average grain size between each sample within the XY and YZ planes diminished after rotary swaging. The lowest average grain size was observed in the centre of the CON + RS sample as well as in the centre of the CON sample. The same trend was observed for the highest average grain size (at the border of the SLM + RS and SLM sample). Based on the obtained data, it can be concluded that the average grain size corresponds with the microhardness. The wavy trend has not been observed for the SLM sample any longer as well as the difference between the two regions (the border and the centre of the sample) in the CON sample. In other words, the hot rotary swaging post-process caused a balance in the microhardness and created a more homogeneous structure. The positive effects of rotary swaging on structure homogenization and grain refinement were proven before also for other metallic materials fabricated via conventional casting (e.g., [71,72,73]), as well as for powder-based steels (e.g., [74,75,76,77]).

As can be seen, the SLM + RS sample featured a significant decrease in HAGB in the horizontal XY plane from 86.2% to 73.3% (see Figure 9a) at the border as well as from 79.6% to 66.9% (see Figure 9b) in the centre of the sample. This was a decrease of 12.9% at the border and 13% in the centre of the sample. Considering the vertical YZ plane a similar course in HAGB fraction was observed, i.e., from 80.6% to 63.4% (a decrease of 17.2%, see Figure 9d) at the border and from 79.5% to 76.5% (a decrease of 3%, see Figure 9e) in the centre. The CON + RS sample exhibited a similar trend of HAGB fraction, which decreased at the border and in the centre of the sample within both planes.

The geometrically necessary dislocation (GND) density maps (depicted in Figure 10) were evaluated based on the OIMs corresponding with the SLM + RS sample presented in the horizontal XY and vertical YZ directions. GNDs have been mapped in a field of 100 × 100 µm around grain boundaries affected by the hot rotary swaging post-process. Each image presents the colour-scale, which illustrates the estimated amount of dislocation density calculated in a number of dislocations·m^−2^, contained within each measured pixel on the map. There was no significant observed difference in GND comparing the SLM and CON samples and the sample subjected to the post-process (SLM + RS and CON + RS). Due to this reason only the SLM and SLM + RS samples are shown in this paper. The GND density increased after rotary swaging and concentrated predominantly on the grain boundaries (see Figure 10a,b). The GND density featured no difference regarding both the areas, the centre and the border within the SLM + RS sample depicted in the horizontal plane (the summary of the GND density is presented in Figure 10c), but, on the contrary, there was a slight difference between the border and the centre area within the sample in the vertical direction. It can be summarized that the post-process imparted an increase in the number of dislocations corresponding with severe plastic deformation. 

Based on the acquired data, the changes in the microhardness and high cycle fatigue were most probably influenced by the following factors: severe plastic deformation (corresponding with the increase in the GND density observed on the grain boundaries), grain size, and HAGB fraction. The plastic deformation, which was imposed by the hot rotary swaging, caused structure changes resulting in a higher microhardness (with no visible wavy trend), a lower average grain size, and an increase in HAGB fraction and GND density. The differences in the microhardness, grain size, and HAGB fraction diminished in the horizontal and vertical direction after hot rotary swaging and exhibited a more homogeneous behaviour. It can be summarized that the post-process of hot rotary swaging featured positive aspects regarding residual porosity and microhardness of the samples. The porosity plays a significant role in fatigue resistance. Parts that have to undergo high-cycle dynamic loading need to withstand a long lifespan. The porosity can impart crack nucleation, which can lead to a lower life of the parts exposed to dynamic loading. In other words, the hot rotary swaging post process has proven to become a functional way to decrease the final porosity corresponding with a higher cycle to failure. The SLM + RS sample exhibited the highest fatigue resistance under the stress of 540 MPa compared to the CON + RS sample. Generally, the samples subjected to the hot rotary swaging post-process exhibited a higher cycle to failure in comparison to other current results, which confirmed the hot rotary swaging as a proper post-processing treatment. 

## 4. Conclusions

This study deals with the AISI 316L stainless steel prepared by a conventional and SLM process. The aim was to investigate the influence of the hot rotary swaging on microhardness, microstructure, and high cycle fatigue. The main results can be described as follows. 

The study has introduced awareness about the hot deformation behavior of the investigated steel with regard to its state after the 3D-printing procedure. The material constants of the Hensel-Spittel formula adapted to reflect this specific state can be then used to predict the corresponding hot flow stress evolution and/or to extend the material database of the FEM simulation software.

The microhardness after the hot rotary swaging under a temperature of 900 °C increased to 310 ± 5 HV and 311 ± 5 HV in the horizontal and vertical directions and exhibited a homogenous trend within the sample. A high-cycle fatigue test resulted in very similar results under the stress of 460MPa, but when the rotary swaged samples were subjected to higher stress (540 MPa), the SLM samples exhibited a higher cycle to failure compared to the conventionally produced samples. The rupture surface was characterized by a complicated viscous relief of the fracture with fine dimples.

The effect of swaging contributed to the significant decrease in the average grain size from 24.4 µm to 3.9 µm for the SLM sample and from 23.40 µm to 4.10 µm for the conventional sample. The effect of rotary swaging contributed to the structure and microhardness changes accompanied by an increase in the geometrically necessary dislocation density on the grain boundaries in the samples. Based on the results it can be concluded that AISI 316L produced via SLM subjected to the post process can be used for dynamically loaded parts. 

## Figures and Tables

**Figure 1 materials-16-03400-f001:**
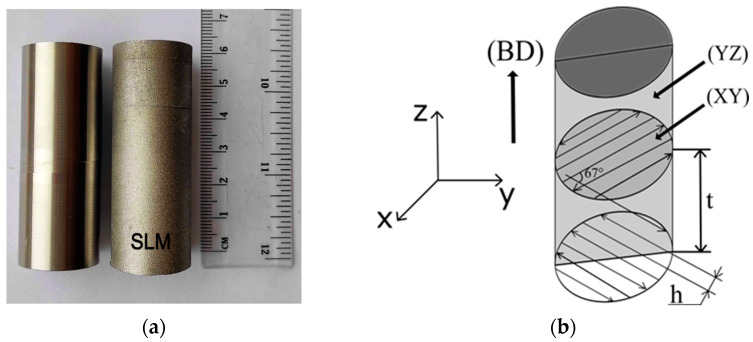
(**a**) Samples of AISI 316L steels produced via SLM and a conventional manner, (**b**) shows the cross-sections used for microscopic analysis.

**Figure 2 materials-16-03400-f002:**
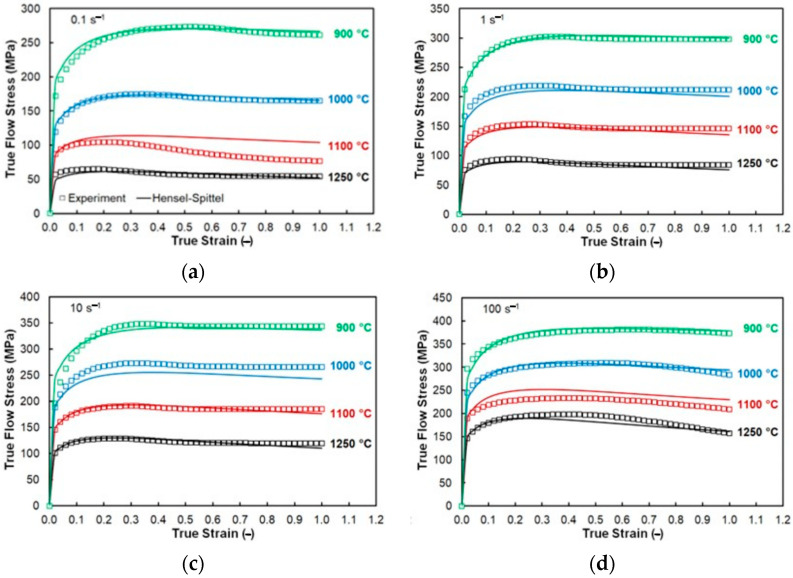
Comparison between the experimental and calculated dataset of AISI 316L steel prepared by 3D printing. (**a**) flow stress courses with the strain rate of 0.1 s^−1^ (**b**) flow stress courses with the strain rate of 1 s^−1^ (**c**) flow stress courses with the strain rate of 10 s^−1^ (**d**) flow stress courses with the strain rate of 100 s^−1^.

**Figure 3 materials-16-03400-f003:**
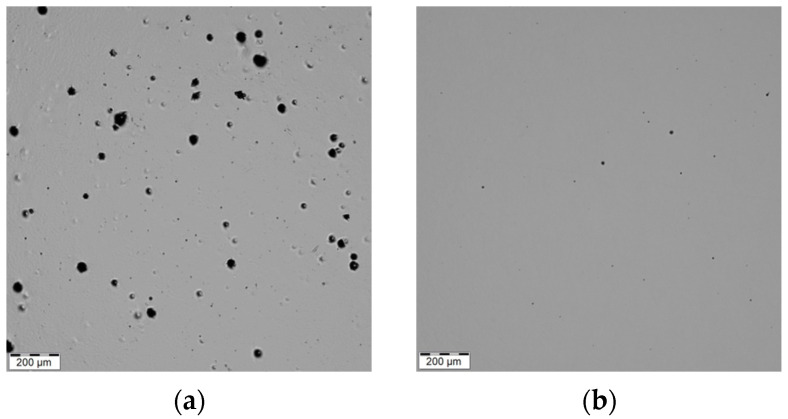
Light optical microscopy (LOM) images show the porosity of the SLM sample (**a**) and SLM + RS sample with reduced number of pores (**b**).

**Figure 4 materials-16-03400-f004:**
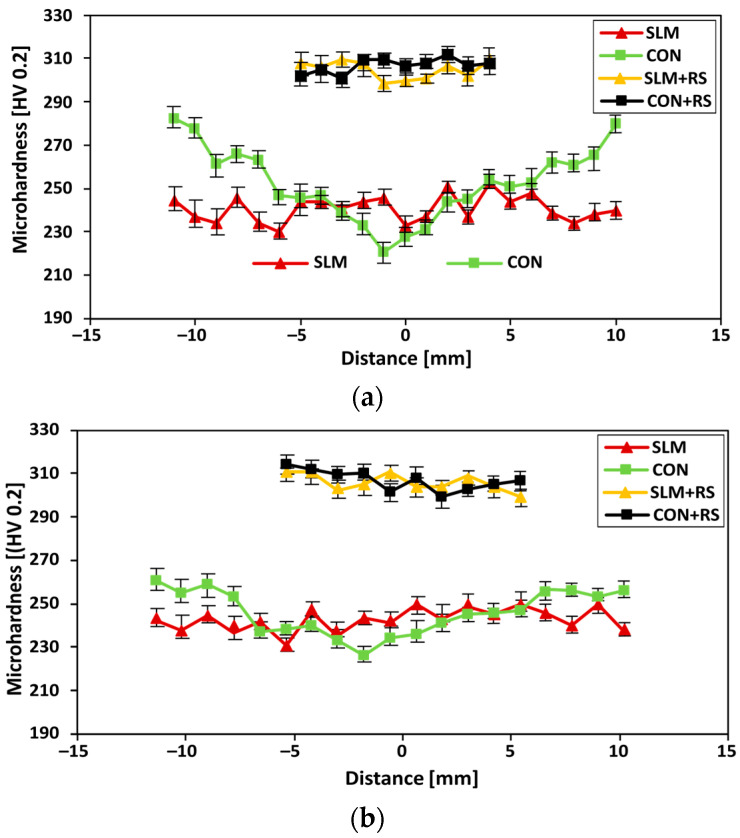
Results of the microhardness HV 0.2 measurements for: (**a**) a distance across the samples in the XY plane, and (**b**) a distance across the samples in the YZ plane.

**Figure 5 materials-16-03400-f005:**
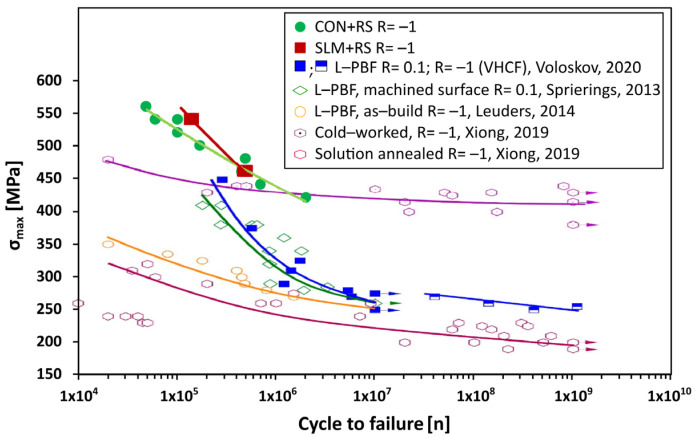
Test of the high cycle fatigue for SLM + RS and CON + RS samples. The figure also includes experimental data acquired by Voloskov et al. [65], Spierings et al. [66], Leuders et al. [67], and Xiong et al. [68]. The points indicated by the arrows correspond to specimens that were not disrupted during the tests.

**Figure 6 materials-16-03400-f006:**
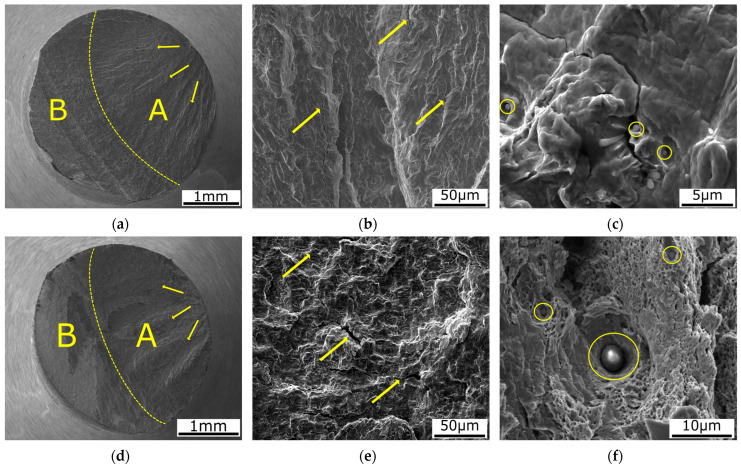
SEM images of the fracture surface with marked propagation (yellow arrows) of the CON + RS2 sample after loading of 540 MPa in the high cycle fatigue regime with unmelted parts (yellow circle) (**a**–**c**), images of the fracture surface of the SLM + RS4 specimen after loading of 540 MPa in the high cycle fatigue regime (**d**–**f**).

**Figure 7 materials-16-03400-f007:**
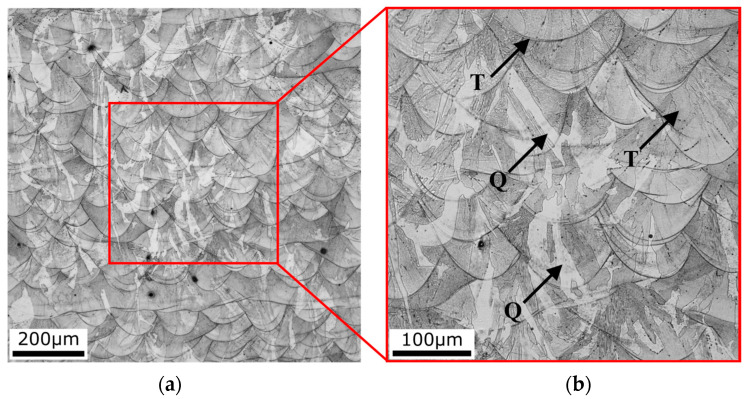
LOM images of the melting pool structure of the SLM sample (**a**) and the detail area (marked with the red square) of the tempered (T) and quenched (Q) regions (**b**).

**Figure 8 materials-16-03400-f008:**
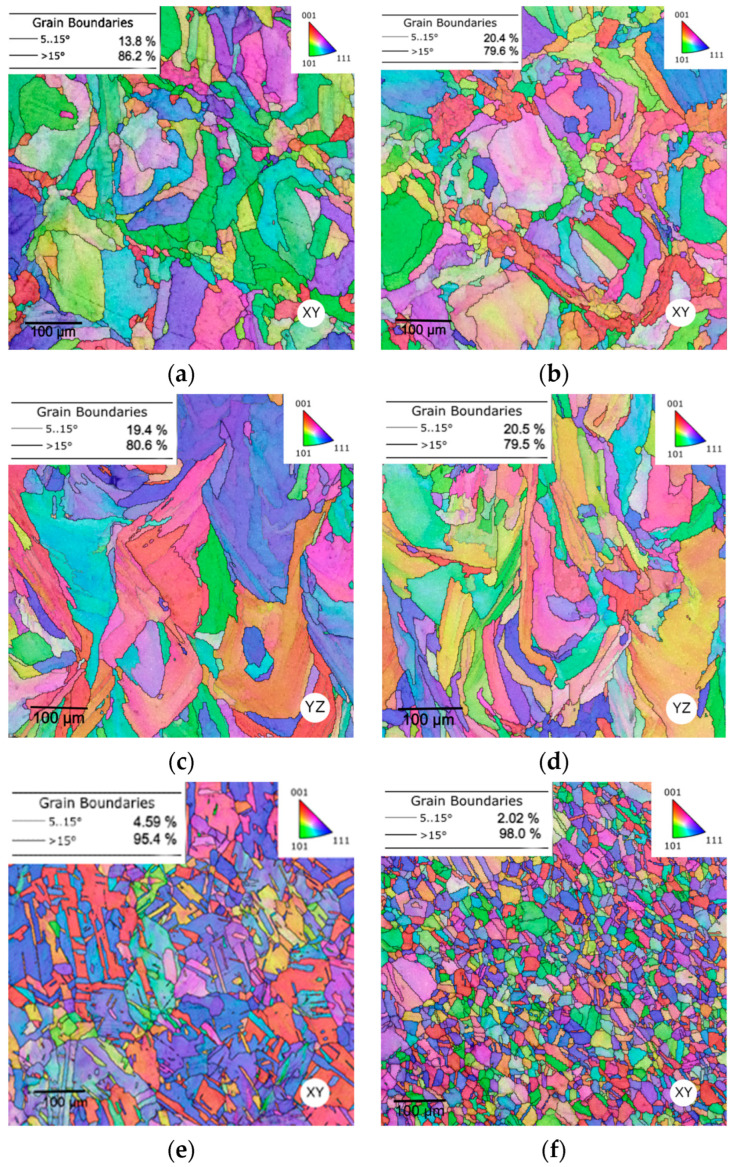
EBSD orientation image maps of the SLM and CON samples: (**a**) a border of the SLM sample and (**b**) a centre of the SLM sample in the XY plane, (**c**) a border of the SLM sample and (**d**) a centre of the SLM sample in the YZ plane, (**e**) a border of the CON sample and (**f**) a centre of the CON sample in the XY plane, (**g**) a border of the CON sample and (**h**) a centre of the CON sample in the YZ plane (**i**) a summary of all states of the average grain sizes in the XY and YZ planes.

**Figure 9 materials-16-03400-f009:**
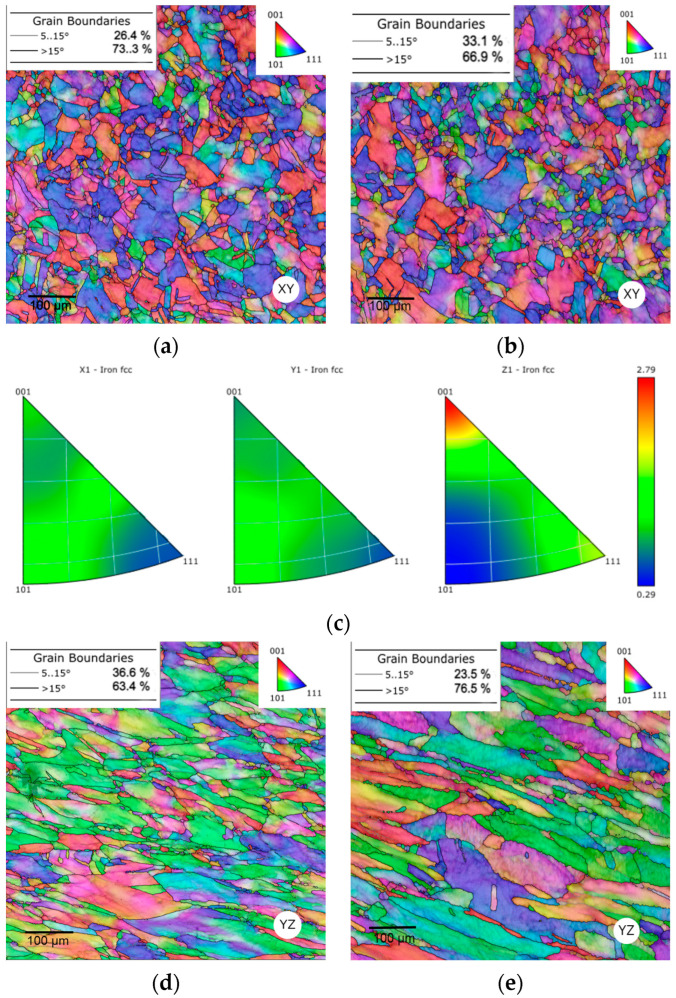
EBSD orientation image maps of the SLM + RS and CON + RS samples: (**a**) a border of the SLM + RS sample and (**b**) a centre of the SLM + RS sample in the XY plane (**c**) IPFs a border of the SLM + RS in the XY plane (**d**) a border of the SLM + RS sample and (**e**) a centre of the SLM + RS sample in the YZ plane (**f**) IPFs a border of the SLM + RS in the YZ plane (**g**) a border of the CON + RS sample and (**h**) a centre of the CON + RS sample in the XY plane (**i**) a border of the CON + RS sample and (**j**) a centre of the sample in the XY plane (**k**) a summary of all states of the average grain sizes in the XY and YZ planes.

**Figure 10 materials-16-03400-f010:**
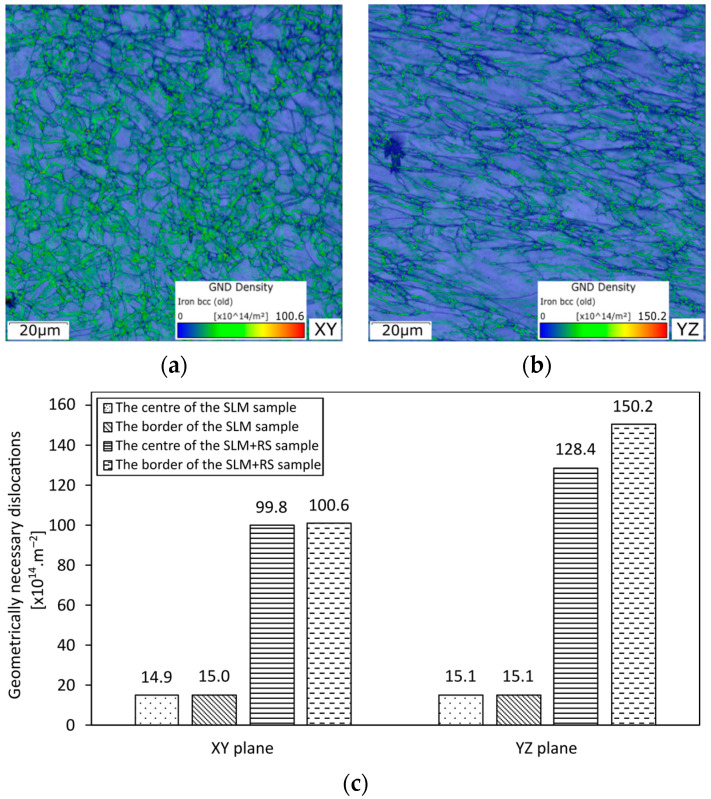
GND density corresponding with the SLM + RS sample (**a**) a border of the sample, the XY plane, (**b**) a border of the sample, the YZ plane, and (**c**) a summary of GND for the border and the centre of the SLM + RS sample in the XY and YZ plane.

**Table 1 materials-16-03400-t001:** Chemical composition of the powder used for the fabrication.

Elements	C	Si	Mn	P	N	S	Cr	Mo	Ni	Fe
Wt. %	Max.	Max.	Max.	Max.	Max.	Max.	Min.–Max.	Min.–Max.	Min.–Max.	Balance
	0.03	1.00	2.00	0.045	0.10	0.03	16.00–18.00	2.00–3.00	10.00–14.00	

**Table 2 materials-16-03400-t002:** Process parameters used for the fabrication of the samples.

Energy Density (J·mm^−3^)	Laser Power (W)	Scanning Speed (mm·s^−1^)	Laser Thickness (mm)	Hatch Spacing (mm)
103	200	650	0.05	0.06

**Table 3 materials-16-03400-t003:** Experimental design of sample states used for the analysis.

Sample	SLM	CON	SLM + RS	CON + RS
Description	as-build	conventional casting	as-build + rotary swaging	conventional casting + rotary swaging

**Table 4 materials-16-03400-t004:** Constants of the Hensel-Spittel formula for AISI 316L steel prepared by 3D printing.

*A*	*m* _1_	*m* _2_	*m* _3_	*m* _4_	*m* _5_	*m* _7_	*m* _8_	*m* _9_
1,094,563	−0.00242	0.20026	−0.23882	0.00091	−0.00105	0.23566	0.00032	−0.82446

**Table 5 materials-16-03400-t005:** Acquired data from the high cycle fatigue test corresponding with the conventionally and SLM post-processed samples.

Sample	CON + RS1	CON + RS2	CON + RS3	CON + RS4	SLM + RS1	SLM + RS2	SLM + RS3	SLM + RS4
Applied load (MPa)	540	540	460	460	540	540	460	460
Cycles to fracture (n)	87,594	60,312	460,047	462,479	146,542	138,846	465,374	469,289

## Data Availability

The original data supporting the research is not publicly available but the data that is not confidential is available on request from the corresponding author.

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
