# Peer review of "High Cycle Fatigue Behaviour of 316L Stainless Steel Produced via Selective Laser Melting Method and Post Processed by Hot Rotary Swaging"

_materials, 2023, doi:10.3390/ma16093400_

Round 1

Reviewer 1 Report

The paper presents a comparison between conventionally produced AISI 316L stainless steel and the same alloy produced via additive manufacturing (SLM). In both cases, the materials are post-processed via hot rotary swaging. Afterwards, they are tested in the high cycle fatigue range. The analysis includes examinations of the microstructure via EBSD. Grain sizes, texture, high-angle grain boundaries, and dislocation density are discussed.

The paper starts with a well-written introduction. Materials and methods are then described in detail, which is followed by a detailed discussion of the results. Novel results are presented. The language level is good. I still recommand thorough proofreading to eliminate minor language errors. Please consider the following comments for a minor revision:

Eq. (2): I assume the variable e refers to the exponential function. Please do not use italics for this ,,variable''. In fact, it's not a variable, but a constant, and using italics here is confusing. Does this equation hold for compression, i.e., negative stresses?

Sect. 2.4: Name the stress levels used in the fatigue tests.

Fig. 3: The results refer to compression tests, i.e., negative stresses and strains. Consequently, the axis should be labelled with absolute stress and strain values.

Fig. 6: change the number format of the labels on the x axis to 1E+... for all numbers. Otherwise, it is too hard to read.

Line 297: Here, you mention that the specimens have been polished before fatigue testing. This is not mentioned in detail in Sect. 2. Since surface quality is crucial in HCF, please provide further details on the surface treatment.

Fig 9 and 10: Add a short text to all images indicating to which specimen the current image refers.

Author Response

Please see the attached the file below.

Reviewer 2 Report

The manuscript created a comparison between Additively prepared SLM and conventionally prepared 316L based on the porosity, hardness, fatigue, and microstructure:

1-A simple comparison is missing in the abstract between these two after rotary swaging, which one is good and how much percentage or change was observed for hardness, priorities, and fatigue?

2- Is HIP or rotary swaging the same treatment or post-treatment processes? why have they written in combined form?

3-What is rotary swaging, and why is it advantageous over the other post-treatment processes for SLM? I didn't find any good reason and it was not even integrated with proper research articles to support the introduction. 

4- What do authors need to achieve after post-treatment of 316L even after selecting the costly SLM Process for better mechanical behavior in the in-built conditions?

5- Is hot compression testing performed before the hot rotary swaging? If not adjust the headings 2.2 and 2.3.

 6- Kindly label Fig. 2, where is the specimens, fixture, and dies.

7- Which testing standard was used for testing in heading 2.4, along with the specimens standard?

8- Fig. 3.2 shows only the porosity comparison of SLM and SLM R+, however, the actual comparison is between the SLM and Conventional after post-treatment.

Author Response

Please see the attached the file below.

Reviewer 3 Report

The abstract is written and arranged very simply and superficially and does not present the work well. In the event that the abstract should be written more attractively. Also, the novelty of the article should be presented clearly. In addition, the conducted tests and their results should be added quantitatively and qualitatively. Keywords can also be modified. Also, the article needs general writing and grammar editing.

The introduction needs to be reformed and deepened. Use the following resources to complete this section. (Effect of welding thermal treatment on the microstructure and mechanical properties of nickel-based superalloy fabricated by selective laser melting, Investigation of welding crack in micro laser welded NiTiNb shape memory alloy and Ti6Al4V alloy dissimilar metals joints).

On what basis were the printing parameters selected? Figure 2.1 shows two rods, while the caption is only for the printed samples. What is the reason for choosing hot compression test temperatures? Why are higher temperatures not used? What is the working temperature of this alloy? The format of the tables must be the same. Table 3 should be corrected. Figure 2 can be deleted.

Requirements must be met to check the fatigue of the printed samples. The following sources should be used in the introduction and research method section to present the fatigue test requirements (Probabilistic framework for fatigue life assessment of notched components under size effects, Defect tolerant fatigue assessment of AM materials: Size effect and probabilistic prospects).

How are the reproducibility of microhardness and fatigue characteristics test results checked? How many times has the test been repeated for each sample? Add an error bar to the results (Figures 5 and 9 and Table 5).

The results section is well organized and categorized. But some parts of it are just reporting the results, which require corrections and deepening the analysis and discussion. Use these sources to analyze the results (Hydrogen embrittlement behavior of SUS301L-MT stainless steel laser-arc hybrid welded joint localized zones, Effects of post-weld heat treatment on the microstructure and mechanical properties of laser-welded NiTi/304SS joint with Ni filler). It is suggested to modify the conclusion section as well as the abstract.

Author Response

Please see the attached the file below.
